# Trans-Kingdom sRNA Silencing in *Sclerotinia sclerotiorum* for Crop Fungal Disease Management

**DOI:** 10.3390/pathogens14040398

**Published:** 2025-04-21

**Authors:** Yuqing Ouyang, Yunong Xia, Xianyu Tang, Lei Qin, Shitou Xia

**Affiliations:** 1Hunan Provincial Key Laboratory of Phytohormones and Growth Development, College of Bioscience and Biotechnology, Hunan Agricultural University, Changsha 410128, China; yuqingouyang95@gmail.com (Y.O.); yun0623@stu.hunau.edu.cn (Y.X.); huxuantingnasha@stu.hunau.edu.cn (X.T.); 2Crop Research Institute, Hunan Academy of Agricultural Sciences, Changsha 410125, China

**Keywords:** trans-kingdom sRNA, *Sclerotinia sclerotiorum*, HIGS, SIGS, VIGS

## Abstract

*Sclerotinia sclerotiorum* is a globally widespread and vast destructive plant pathogenic fungus that causes significant yield losses in crops. Due to the lack of effective resistant germplasm resources, the control of diseases caused by *S. sclerotiorum* largely relies on chemical fungicides. However, excessive use of these chemicals not only causes environmental concerns but also leads to the increased development of resistance in *S. sclerotiorum*. In contrast, trans-kingdom sRNA silencing-based technologies, such as host-induced gene silencing (HIGS) and spray-induced gene silencing (SIGS), offer novel, effective, and environmentally friendly methods for the management of *S. sclerotiorum* infection. This review summarizes recent advances in the identification of *S. sclerotiorum* pathogenic genes, target gene selection, categories, and application of trans-kingdom RNA interference (RNAi) technologies targeting this pathogen. Although some challenges, including off-target effects and the efficiency of external sRNA uptake, exist, recent findings have proposed solutions for further improvement. Combined with the latest developments in CRISPR/Cas gene editing and other technologies, trans-kingdom RNAi has significant potential to become a crucial tool in the control of sclerotinia stem rot (SSR), mitigating the impact of *S. sclerotiorum* on crop production.

## 1. Introduction

Crop production often faces severe threats from various fungal diseases, among which sclerotinia stem rot (SSR), caused by *Sclerotinia sclerotiorum*, has become one of the most challenging due to its wide host range and destructive nature [1,2]. SSR not only causes severe production loss in rapeseed, but also brings about significant economic losses all over the world, particularly in China [3], the United States [4], Canada, Australia, and Brazil [5]. Moreover, this disease leads to reduced oil content, altered fatty acid composition, and the production of mycotoxins, which severely affect the quality of rapeseed [6]. *S. sclerotiorum* spreads via ascospores in soil or air and secretes oxalic acid and cell wall degrading enzymes to destroy plant tissues and induce rot and even plant death [7]. The dormant sclerotium can survive in the soil for extended periods, further complicating the efforts of disease control [8]. Due to the lack of high-resistant varieties to *S. sclerotiorum*, the effectiveness of traditional breeding approaches is limited presently for disease resistance improvement [9]. Currently, chemical fungicides remain the most effective for controlling SSR, but their continuous use not only enhances the resistance in *S. sclerotiorum* but also causes environmental issues, posing a threat to human health [10]. Therefore, green, efficient, and host-specific strategies for SSR management have become of vital importance in recent years.

Since the discovery of the conserved RNA interference (RNAi) mechanism mediated by small RNAs (sRNAs) in eukaryotes [11] and its cross-species delivery phenomenon, trans-kingdom RNAi has gradually emerged as a potential solution for plant protection [12,13,14], among which host-induced gene silencing (HIGS), as one of the trans-kingdom RNAi technologies, utilizes genetic engineering to enable host plants to specifically express double-stranded RNA (dsRNA) or sRNAs targeting pathogen genes, which are then transported across kingdoms to silence the virulent genes in pathogen [15]. The plant’s inherent defense mechanisms against viruses make the virus a potential tool for producing and transmitting sRNAs, thus enabling another form of RNAi known as virus-induced gene silencing (VIGS) [16,17]. Recent studies have further revealed that certain pathogenic fungi can actively uptake sRNAs from their environment [18,19,20]. This discovery of environmental RNAi has led to the development of spray-induced gene silencing (SIGS), involving the exogenous spraying of synthetic dsRNA to control disease [21,22].

## 2. Virulence-Related Genes Newly Identified in *S. sclerotiorum*

*S. sclerotiorum* was previously considered a necrotrophic pathogen, but subsequent studies have shown that two stages exist during its infection: it suppresses plant immune responses by effectors and other virulence factors to ensure successful colonization in the early stage and triggers plant tissue necrosis by producing oxalic acid, reactive oxygen species (ROS), CWDE, and toxins, accelerating the infection and obtaining nutrients in the later stage. Therefore, *S. sclerotiorum* is more like a hemibiotrophic pathogen similar to *Magnaporthe oryzae* [7,23,24]. Many researchers have explored the key regulatory genes involved in the growth, development, and pathogenicity of *S. sclerotiorum* at different stages, as well as the interactions between pathogen and host, e.g., the effector-encoding gene *SsITL*, which regulates the JA/ET signalling pathway [25]; the NEP1-like proteins (NLPs) encoding genes *SsNEP1* and *SsNEP2*, first identified in 2010 [26], which induce host plant cell death; and the *oah* gene, first discovered in 2015, which is associated with oxalic acid synthesis [27], as reviewed by Derbyshire and Zhu [28,29]. Nevertheless, the consistent analysis of the *S. sclerotiorum* genome and the advancement of gene function research technology in recent years have provided a wider range of sources and new ideas for the study and prevention of *S. sclerotiorum* growth mechanism. Thus, we focus primarily on the latest research findings from the past three years (Table 1) in this review.

Some new genes related to transcriptional reprogramming and signalling pathways were found recently to regulate the virulence of *S. sclerotiorum*. APSES family transcription factors *Ss*StuA [30] and Zn2Cys6-type transcription factor *Ss*ZNC1 [31] were found to drive virulence gene expression and osmotic stress response pathways, respectively, regulating sclerotium formation and infection structure differentiation. *SsATG1*, *SsATG8*, and *SsATG13* were proved to be autophagy-related genes, which are indispensable for the virulence of *S. sclerotiorum* [32,33], and the FOX family transcription factor *Ss*FoxE3 interacts with the promoter of *SsATG8*, activating its transcription to regulate autophagy, promote appressoria development, and enhance ROS scavenging ability, ultimately affecting the pathogenicity of *S. sclerotiorum* [34]. The cAMP pathway-dependent catalytic subunit *Ss*PKA2 and regulatory subunit *Ss*PKAR regulate the pathogenicity of *S. sclerotiorum* by affecting the autophagy regulation of glycerol accumulation in hyphae [35]. The MAPK cascade pathway is also essential for the virulence of *S. sclerotiorum*. Its important components *Ss*te50, *Ss*Ste11, *Ss*Ste7, and *Ss*Smk1 regulate hyphal fusion and host penetration together with its downstream transcription factor *Ss*Ste12 [36]. Several studies revealed the contribution of the *Ss*Os4-*Ss*Os5-*Ss*Hog1 cascade in the high osmotic glycerol (HOG) pathway [37] and its upstream regulatory gene *SsShk1* to the pathogenicity of *S. sclerotiorum* [38].

New effector proteins were also found to inhibit the plant’s immune response. *Ss*PEIE1 binds to the Arabidopsis plasma membrane *At*HIR4, the key immune factor, inhibiting its oligomerization-mediated hypersensitive response (HR), weakening the host’s early defence barrier [39]. The chloroplast-targeted effector *Ss*CTP1 interacts with the plant’s conserved enzymes *Gm*CPX and *Gm*SKL2, disrupting chitin-triggered immune signalling and enabling the pathogen’s cross-host adaptation [40]. The lectin effector protein *Ss*CVNH binds to the class III peroxidase *At*PRX71, inhibiting ROS burst and defence gene activation, thereby systematically enhancing host susceptibility [41]. The salicylic acid hydroxylase *Ss*Shy1 weakens defence signals by metabolizing host salicylic acid [42]. Some secreted necrosis-inducing proteins are also essential for the virulence of *S. sclerotiorum*. The xylanase *Ss*Xyl2 acts as a cell death-inducing protein, triggering HR in the host by binding to *Nb*HIR2, thereby promoting pathogen colonization [43]. The glycosylphosphatidylinositol-anchored protein *Ss*Gsr1 induces host cell death and activates PTI, forming an “immune double-edged sword” strategy [44]. Necrosis and ethylene-induced peptide 2 (*Ss*NEP2) are essential for the virulence of *S. sclerotiorum*, but its conserved NLP20 triggers host mode-triggered immunity, thereby promoting the necrotrophic lifestyle of *S. sclerotiorum* [45]. The cutinase *Ss*Cut1 enhances enzyme activity to facilitate host epidermal penetration and trigger defense responses in host plants, contributing to the virulence of *S. sclerotiorum* [46].

For other growth and virulence regulators, the key enzyme in chitin–glucan metabolism *Ss*AGM1, which catalyses UDP-GlcNAc synthesis, is critical for sclerotium formation and virulence [47]. The nitrate reductase *Ss*NR coordinates infection cushion development and oxalic acid secretion by regulating nitrogen metabolism and ROS detoxification genes (e.g., *SsCyp*) [48]. The secretory pathway components *Ss*Emp24 and *Ss*Erv25 act as cargo receptors mediating effector protein transport, promoting acid accumulation and infection cushion formation [49]. The SWI/SNF complex component *Ss*Snf5 manipulates sclerotium formation, and the stress response module *Ss*Snf5-*Ss*Hsf1 mediates the formation and maturation of infection cushions, as well as the detoxification of excessive ROS, subsequently regulates the virulence [50]. Additionally, genes such as *SsCak1* [51], *SsMRT4* [52], *SsZfh1* [53], *SsTrx1* [54], and *SsArf6* [55], due to their vital roles in the regulation of growth, development, and virulence, are also potential candidate targets for controlling *S. sclerotiorum* through trans-kingdom RNAi methods.
pathogens-14-00398-t001_Table 1Table 1Virulence-related gene recently cloned in *Sclerotinia sclerotiorum*.Coding GeneTypologyFunctionPathwayRelated Gene/ProteinReference*SsPEIE1*Plant early immunosuppressive effectorPositively meditate virulenceInhibit oligomerization-mediated immune responsesHypersensitive-induced reaction (HIR)[39]*SsStuA*APSES family transcription factorInvolved in vegetative growth, sclerotia formation, fungicide tolerance, and full virulence CWI pathway and ROS response*Ss*Smk3, *SsAGM1*[30]*SsZNC1*Fungal-specific Zn2Cys6 transcription factorsInvolved in virulence, sclerotial development, and stress responseRegulate the expressions of genes related to metabolic pathways, biosynthetic pathways, secreted proteins, and autophagy/[31]*SsFoxE3*Transcription factorContribute to sclerotia, compound appressoria formation, and pathogenicity
Promoter of *Ss*Atg8[34]*Sszfh1*C2H2 transcription factorContribute to the proper development and maturation of sclerotia and apothecia, hyphae ROS production and melanin accumulation*Ss*NOX1 and *Ss*NOX2[53]*SsFkh1*Transcription factorHyphal development, virulence, sclerotia formation, and maintenance of the cell wall integrity (CWI)MAPK signaling pathway*Ss*Mkk1[56]*SsSnf5*Transcription factorMaturation of infection cushions, tolerance of stress, and pathogenicity*Ss*Snf5-*Ss*Hsf1-*Ss*Hsp70 module*Ss*Hsf1, *Ss*Hsp70[50]*SsSte12*Transcription factorSclerotia formation, compound appressoria development and sexual reproductionPheromone response MAPK cascade/[36]*SsCTP1*Effector proteinInhibit plant immunity and promote pathogen infectionsTargeting the chloroplast proteins *Gm*CPX and *Gm*SKL2Coproporphyrinogen-III oxidase (GmCPX) and shikimate kinase 2 (GmSKL2)[40]*SsCVNH*Effector proteinInhibit the plant immune responseInterfere with ROS homeostasis, reduce peroxidase activity, and decrease the activation of defence gene transcription, thereby contributing to the virulence of *S. sclerotiorum*Class III peroxidase *At*PRX71[41]*SsINE1/5*Effector proteinInduce cell death/NLR protein[57]*SsYCP1*Effector proteinPathogenicity//[58]*SsEmp24/SsErv25*Secreted protein Positively regulate vegetative growth and sclerotium formation, cushion formation and virulence*Ss*Emp24 interacts with *Ss*Erv25 to form a p24 protein complex and act as cargo receptors to accept and carry specific cargo proteins between ER and Golgi apparatus, including secreted proteins, GPI-APs, and membrane proteinsInteract with each other[49]*SsERP1*Secretory proteinPathogenicityPlant ethylene signaling pathway/[59]*SsCut1*Cutinsaesecreted proteinPromote virulence without the influence on growth rate, colony morphology, oxalic acid production, infection cushion formation, and sclerotial developmentEnhance the cutinase activity and activate plant immune response/[46]*SsTrx1*:Substrates for reductive enzymesRegulate hyphal growth rate, mycelial morphology and sclerotial development, promote pathogenicity and oxidative stress toleranceTrx system, ROS scavenging pathway
[54]*AGM1*N-acetylglucosamine-phosphate mutaseFunction in vegetative growth, sclerotia production, and infection cushion involved in the response to osmotic stress and inhibitors of cell wall synthesisCatalyzes intramolecular phosphoryl transfer on the phosphosugar substrate GlcNAc-6P in the biosynthesis of UDP-GlcNAc, therefore meditating the synthesize of chitin, an important component of cell walls/[47]*SsCak1*Protein kinasePositively contribute to the growth, development, and pathogenicity Effectuating the phosphorylation of CDC28/[51]*SsOs4*MPKKK (phosphotransferase)Contribute to mycelial growth and differentiation, sclerotia formation, virulence, hyperosmotic adaptation, fungicide sensitivity, and phosphorylation of *Ss*Hog1 The high osmolarity glycerol (HOG) pathway: regulate phosphorylation of *Ss*Hog1 and the expression level of *Sshog1*
*SsglpA* and *Ssfps1*, *Sshog1*[37]*SsNR*Nitrate reductaseContribute to mycelia growth, sclerotia formation, infection cushion formation, cell wall and membrane integrity, OA production, and virulenceNitrogen metabolism*CYP* and infection cushion development-related genes (*Ggt1*, *Sac1*, and *Smk3*)[48]*SsPDE2*cAMP phosphodiesteraseSclerotia formation, oxalic acid accumulation, infection cushion functionality, and virulencecAMP-dependent inhibition of MAPK signaling/[60]*SsAtg1*Protein kinaseNutrient utilization, sclerotia development, cell wall integrity, and pathogenicity Autophagy pathway/[33]*SsShy1*Salicylate hydroxylaseHyphae growth and sclerotia production, cell wall integrity, and pathogenicityTricarboxylic acid cycle of glucose metabolism/[42]*SsDim5*Histone methyltransferaseCell integrity, oxidative stress and osmotic stress response, and pathogenicityH3K9 trimethylation, mycotoxins biosynthesis/[61]*SsHog1*Histidine kinaseOsmotic adaptation, fungicide sensitivity, and virulence High osmolarity glycerol (Hog1) stress response, MAPK signal transduction pathwayShk1[38]*SsXyl2*Glycosyl hydrolase family 11 xylanaseHost cell physiology, pathogenicity/Hypersensitive-induced reaction protein 2 (*Nb*HIR2)[43]*SsCat2*CatalaseContribute to the predominant catalase activity, H_2_O_2_-induced oxidative stress response, cell integrity of hyphae, and pathogenicity/*AOX* gene[62]*Ssgar1*,Plant cell wall degrading enzymes (CWDEs)Virulence, tolerance to salt stress., and cell wall stability D-galacturonic acidcatabolism/[63]*SsGH5*Cell wall degrading enzymesGlucan utilization, plant cell death, and pathogenicity//[64]*SsSte11*MAPKKKSclerotia formation, hyphal fusion, vegetative growth, and pathogenicity*Ss*Ste50-*Ss*Ste11-*Ss*Ste7-Smk1 cascade*Ss*Ste50, *Ss*Ste7[36]*SsSte7*MAPKKHyphal fusion and pathogenicity*Ss*Ste50-*Ss*Ste11-*Ss*Ste7-Smk1 cascade*Ss*Ste1, Smk1[36]*Smk1*MAPKHyphal fusion and pathogenicity*Ss*Ste50-*Ss*Ste11-*Ss*Ste7-Smk1 cascade*Ss*Ste7[36]*SsSte50*Adaptor proteinNormal vegetative growth, hyphal fusion, sclerotia formation, compound appressoria formation, and pathogenicity*Ss*Ste50-*Ss*Ste11-*Ss*Ste7-Smk1 cascade*Ss*Ste11[36]*SsATG8*Ubiquitin-like proteinVegetative growth, sclerotial formation, oxalic acid (OA) production, compound appressoria development, and virulenceUbiquitin–proteasome and selective autophagy pathways*Ss*NBR1[32]*SsGsr1*Glycosylphosphatidylinositol (GPI)-anchored cell wall proteinCell wall integrity of hyphae, pathogenicity, and cell death//[44]*SsNEP2*Necrosis and ethylene-inducible peptideInvolved in fungal virulence by affecting ROS levels and causing cell death//[45]*Sspka2/SspkaR*Catalytic/regulatoryAutophagy regulation and appressorium developmentcAMP-dependent protein kinase A (PKA) signaling pathway, carbohydrate metabolism and mobilization/[35]*SsMRT4*Ribosome assembly factorContribute to mycelia growth, appressorium formation, oxalic acid production, and abilities to ROS elimination and resistance to oxidative and osmotic stresses//[52]*SSA**S. sclerotiorum* agglutininPositively regulate sclerotial development and resistance to *C. minitans* mycoparasitism but negatively regulate pathogenicity and resistance to chemical stresses//[65]*SsArf6*ADP-ribosylation factorsHyphal growth and development, melanin accumulation, appressorium formation, and fungal virulence//[55]


## 3. Target Gene Selection and Categories of Trans-Kingdom RNAi

### 3.1. Trans-Kingdom RNAi Target Gene Selection

Not all genes that influence the growth, development, and virulence of pathogens can serve as effective and suitable target [66]. The selection of genes with clear function related to pathogenicity, minimal functional redundancy, and few or no endogenous homologs or similar sequence genes in the host and surrounding species can reduce off-target effects and avoid adverse impacts on the host and environment.

The targets of traditional fungicides have shown significant application value on target gene selection in trans-kingdom RNAi. For example, in the sterol biosynthesis pathway, demethylation inhibitors (DMIs), such as tebuconazole, propiconazole, and myclobutanil specifically, inhibit the cytochrome P450-dependent lanosterol 14α-demethylase (CYP51), disrupting the integrity of the fungal cell membrane. By genetically expressing dsRNA targeting the *CYP51* of fungal in Arabidopsis, it is possible to effectively silence three *CYP51* transcript variants in the pathogen, significantly enhancing host’s resistance to *Fusarium graminearum* [67]. In addition to sterol metabolism, fungal microtubule protein biosynthesis system, particularly members of the *β-TUB* gene family, has been confirmed as an effective target for RNAi-based control of fungal diseases [68]. A typical example is the study of benzimidazole fungicides, where RNAi vectors expressing complementary dsRNA of *β2-tubulin* genes have been successfully used to achieve systemic resistance in soybeans to *Colletotrichum truncatum* [69]. These studies provide key molecular targets and feasibility validation for plant protection strategies based on trans-kingdom RNAi technology.

As the formation of appressoria and haustoria is crucial in the early stages of infection, related genes can be potential resources for RNAi constructs designing to suppress fungal growth and pathogenicity. When HIGS targets fungal genes that are preferentially expressed rather than constitutively expressed in haustoria, it can more effectively enhance barley’s resistance to wheat stripe rust [70]. One study screened 86 potential wheat stem rust fungus targets that are preferentially expressed in haustoria and found that 10 of these genes exhibited suppression of disease symptoms, with this suppression correlated with reduced abundance of the target transcripts [71]. The protein kinase A subunit (*Ps*CPK1) was significantly upregulated at the early stage of infection (18 h post-inoculation), and virus-mediated transient silencing of *PsCPK1* resulted in about 50% reduction in appressorium development [72]. Furthermore, *SsCak1* is an important regulatory gene for the formation of appressorium in *S. sclerotiorum*, and using *SsCak1* as a target for HIGS can significantly enhance the host’s resistance to SSR [51].

Effector proteins, as key regulatory elements in plant-pathogen interaction systems, provide important theoretical insights for selecting targets for trans-kingdom RNAi. Recent studies have shown that the wheat powdery mildew fungus (*Blumeria graminis f.* sp. *tritici*) secretes an RNase-like effector protein SvrPm3 a1/f1 that specifically inhibits the recognition function of the Pm3 resistance locus, thus escaping the plant immune system [73]. By genetically engineering plants to express hpRNA targeting *SvrPm3 a1/f1*, researchers successfully silenced the target mRNA that coded effector protein, partially restoring wheat’s resistance to powdery mildew. This provides direct evidence for the target selection of trans-kingdom RNAi. Further studies have revealed that pathogens can interfere with plant defense mechanisms via miRNA-mediated gene regulatory networks. For instance, *Puccinia striiformis f.* sp. *tritici* (*Pst*) secretes the miRNA *Pst-milR1*, which specifically silences the expression of the host β-1,3-glucanase encoding gene *PR2*. The enzyme (EC 3.2.1.39), as a key antifungal protein, defends against the pathogen by hydrolyzing the β-glucan component of the fungal cell wall [74,75]. Notably, targeting the precursor RNA sequence of *PstmiR1* with HIGS technology can effectively block the biosynthesis of this virulence factor and restore the normal expression level of *PR2*. These advances suggest that by systematically analyzing the mechanisms of *S. sclerotiorum* pathogenic effectors, a multi-target coordinated regulation system based on RNAi will significantly improve the efficacy of trans-kingdom RNAi in the management of plant fungal diseases.

Whole genome sequencing also laid a crucial foundation for trans-kingdom RNAi target gene screening [76]. Firstly, genome-wide comparative analysis enables the systematic identification of *S. sclerotiorum*-specific gene sequences. These sequences exhibit no significant homology with host plants or other non-target microorganisms, thereby significantly reducing off-target effects in trans-kingdom RNAi. Secondly, genomic data provide key insights into the secretion mechanisms of *S. sclerotiorum* effectors and the genes associated with their transport. For example, identifying the functions of pathogen-secreted proteins can offer a theoretical basis for designing plant-based delivery systems for dsRNA [77]. Additionally, integrating genome sequencing with transcriptomic data from infection stages allows for the precise identification of highly expressed pathogenicity-related genes during infection. This facilitates the selection of target genes that play a critical role at key infection stages [78].

### 3.2. Categories of Trans-Kingdom RNAi

RNAi is a highly conserved post-transcriptional gene silencing mechanism that mediates sequence-specific mRNA degradation via small RNA molecules and widely presents in eukaryotes [11,79,80]. Its mechanism, which is activated by the dsRNA and processed with the participation and manipulation of the RNA-induced silencing complex (RISC), is widely and clearly interpreted by many researchers [81,82].

To date, three categories of trans-kingdom RNAi technology was developed based on the conservative mechanism of RNAi and the further findings of cross-species RNAi phenomenon [83,84] (Figure 1). In HIGS, pathogen-specific dsRNA targeting pathogen genes are expressed by genetically modified plants and processed into 21–25 nt double-stranded small interfering RNA (siRNA). The siRNA is then transferred from the plant cell into a pathogen cell through external pathogen RNA uptake mechanism. When transmitted into a pathogen cell, the siRNA is unwound by the RNA-induced silencing complex (RISC), and the guide strand RNA is guided to the targeted gene sequence, subsequently mutating the interpretation of the targeted gene, while the passenger strand is specifically degraded [85,86]. Since the HIGS was first confirmed in the plant pathogenic filamentous fungus *Blumeria graminis f.* sp. *tritici* in 2010 [87], it has been successfully applied in several plant–fungal pathosystems, such as cotton-*Verticillium dahliae*, rice-*Magnaporthe oryzae*, and soybean*-Phakopsora pachyrhizi*, showing broad-spectrum disease resistance potential [84,88,89,90].

Nevertheless, HIGS faces biosafety concerns and ethical issues because it involves genetic modification technology and is subject to regulatory and policy restrictions, as well as social impacts, during its promotion and practical production. To overcome the limitations of transgenic technology, SIGS has emerged as a non-genetically modified (non-GM), environmentally friendly strategy. The theoretical basis of SIGS comes from the discovery that a pathogen, such as *Botrytis cinerea*, possesses the ability to absorb environmental RNA. Experimental evidence has shown that exogenously sprayed sRNA/dsRNA formulations targeting key pathogen genes (e.g., *DCL1*/*DCL2* encode the key components of the RNAi machinery, which is also proved to be the pathogenic mechanism in *Botrytis cinerea*) can effectively suppress the development of various fungal diseases in post-harvest agricultural products [19,20]. Notably, dsRNA not only exerts its effects at the treatment site but can also be transported through the plant’s internal transport system, resulting in distal protection, as observed in the systemic resistance induced in the barley-*Fusarium graminearum* system [91]. Recent studies have further confirmed that SIGS can reduce the biomass of *Phakopsora pachyrhizi* by 75% and exhibits significant control effects against pathogens, such as *S. sclerotiorum*, marking the broad application prospects of this technology in crop protection [90].

VIGS, however, is another kind of trans-kingdom RNAi method, which uses genetically engineered viruses or plant cells containing meditated viral genome to evoke RNAi in a host plant. It utilizes a natural defense mechanism in plants against viral infections. By methods such as *Agrobacterium*-mediated agro-infiltration, fresh snap infection, DNA bombardment, and RNA transcript rub inoculation, the virus carrying the constructed target gene sequence is delivered into the plant. This activates the endogenous RNA-guided RNA polymerase in the host cells, which replicates within the plant and produces dsRNA targeting the specific gene, triggering RNA interference in the plant, ultimately silencing the target gene [92]. Since the successful construction of the first tobacco mosaic virus VIGS vector in the 1990s, which silenced the *NbPDS* gene expression in *N. benthamiana* and caused a bleaching phenotype, this gene interference method has been widely used for gene function detection in plants and pathogens [16,93]. With the development of technology, VIGS has recently been proven to effectively control diseases in plants. For example, targeting the chitinase gene of *F. oxysporum* through VIGS can effectively control its pathogenicity and spread in cucumbers [94].

## 4. Application of Trans-Kingdom RNAi in *S. sclerotiorum*

Among the genes identified as related to *S. sclerotiorum* growth, development, and virulence, some have been successfully applied in trans-kingdom RNAi approaches (Table 2). To date, the target genes proved to be applicable for trans-kingdom RNAi in *S. sclerotiorum* have almost covered all stages of its growth, development, and formation of pathogenicity. Increasing results have shown that trans-kingdom RNAi plays an important role in reducing the infectivity, pathogenicity, and lethality of *S. sclerotiorum* during the process of biotrophic and necrotic infection.

Chitin synthase genes (CHS), for instance, are responsible for synthesis of chitin, a polysaccharide that serves as a major structural component of the cell walls in most fungi. Therefore, the chitin biosynthesis pathway has been identified as a crucial target for antifungal drug development [95]. The Andrade research team first demonstrated, using HIGS technology, that RNA interference targeting the *CHS* gene significantly enhanced the resistance of tobacco T1 generation plants to SSR [96]. In the pathogenesis of *S*. *sclerotiorum*, *oxaloacetate hydrolase* (*SsOha1*) is a key gene regulating oxalate biosynthesis, with its expression product being a core virulence factor of the pathogen. Rana et al. successfully constructed a HIGS expression vector targeting *SsOha1* and achieved Arabidopsis genetic transformation. Pathogenicity analysis showed that the T3 generation transgenic lines exhibited significantly increased resistance to *S. sclerotiorum* infection compared to the wild-type controls [97]. Further studies confirmed that the *SsOha1* gene silencing system, based on the bean pod mottle virus (BPMV) vector, effectively reduced the severity of soybean sclerotinia stem rot [98]. The mitogen-activated protein kinase (MAPK) signaling cascade is a key pathway in regulating the virulence of *S. sclerotiorum*. Host-mediated expression of dsRNA targeting genes related to this pathway can significantly impair the pathogen’s virulence to *N*. *benthamiana* and transgenic Arabidopsis [36]. In addition, HIGS intervention strategies targeting a series of genes, such as *SsCak1* [51], *SsTrx1* [54], *SsCnd1* [99], *SsGAP1* [100], *SsPac1*, and *SsSmk1* [101], have shown significant application potential in plant disease control.

Based on the fact that *S. sclerotiorum* can efficiently take up environmental dsRNA through endocytosis [20], SIGS technology has become a new strategy for the prevention and control of SSR. A typical example is the early infection stage of *S. sclerotiorum*, where the secreted protein *Ss*ERP1 is highly expressed. Targeted silencing of its in vitro-synthesized dsRNA can effectively inhibit the pathogen’s infection process [59]. Meanwhile, various studies have shown that SIGS plays an important role in controlling the progression of sclerotinia diseases (Table 2). Notably, hpRNA targeting the *ABHYDROLASE-3* gene in Arabidopsis and rapeseed not only significantly delayed the infection process [102] but also enhanced the host’s resistance when the dsRNA of this gene was sprayed onto the leaves [103]. Even more groundbreaking is the combination of dsRNAs targeting *ABHYDROLASE-3* and three other genes (chitin-binding domain, MAPK, and oxaloacetate hydrolase), which, when sprayed on plants, achieved better disease control than single-gene dsRNA application [104]. In support of this, exogenous application of the *SsVPS51+SsDCTN1+SsSAC1* combined dsRNA formulation to lettuce and collard leaves also showed significant resistance enhancement effects [20]. These findings suggest that the combined application strategy of multi-target gene silencing systems may become an important research direction for optimizing disease control strategies in the future.

To date, there have been no reports on trans-kingdom RNA interference directly induced via viral RNAi vectors in *S. sclerotiorum*. Nevertheless, VIGS is applied in the control of *S. sclerotiorum* by either interfering pathogen resistance-related genes and pathways, consequently inhibiting the function of pathogen’s virulent factors, or by mutating pathogen’s targeted genes in combination of HIGS. For example, the *Ss*INE5 receptor coding gene *NbNLR* in *N. benthamiana* was mutated via VIGS, thereby suppressing the function of *Ss*INE5 [57]; VIGS targeting *GmRBOH-VI* in soybean via bean pod mottle virus (BPMV) enhanced resistance to *S. sclerotiorum* and significantly reduced ROS levels during infection process [105]; *NbSGT1* is required for disease-associated cell death caused by *S. sclerotiorum*, and silencing the expression of *NbSGT1* by VIGS via tobacco rattle virus (TRV) can improve host resistance to *S. sclerotiorum* [106]; TRV was applied as vehicle to induce the expression of *SsCnd1*-RNAi construction in host plant *N. benthamiana,* consequently silencing *SsCnd1* in *S. sclerotiorum* [99].

Interestingly, recent studies have identified various viruses that can directly reduce the virulence of *S. sclerotiorum* by upregulating RNAi-related genes or downregulating virulence factor-related genes, such as the *Ss*DFV3 virus, *S. sclerotiorum* endonucleic acid virus 3 (*Ss*EV3), and the DNA virus *Ss*HADV-1 [107,108,109]. While the mechanisms by which fungal viruses reduce pathogen virulence require further investigation, the genes and pathways targeted by these viruses may serve as important sources for trans-kingdom RNAi control strategies. Given the trans-kingdom nature of viruses in both plants and fungi [110], asymptomatically infecting viruses such as *Ss*EV1 and *Ss*EV2 may also be potential vectors for other trans-kingdom RNAi target genes [111]. Therefore, VIGS, as well as its extension and relative biocontrol agents, are also important for the management of SSR in crops.

Though there have been significant findings on trans-kingdom RNAi application focusing on *S. sclerotiorum* control, many target validation studies have used *A. thaliana* and *N. benthamiana* as host plants, and further verification in more crops is still needed. Additionally, most application effect validations have been conducted under controlled laboratory conditions, and field application results are yet to be further confirmed.
pathogens-14-00398-t002_Table 2Table 2Application of trans-kingdom RNAi target gene.Target GeneApplied MethodHost Reference *SsSnf5-SsHsf1-SsHsp70*HIGS*N. benthamiana*[50]*SsSte50*HIGS*N. benthamiana*, *A. thaliana*[36]*SsERP1*SIGS*Tobacco*[59]*SsTrx1*HIGS*A. thaliana*, *N. benthamiana*[54]*SsCak1*HIGS*N. benthamiana*[51]*SsPDE2*HIGS*N. benthamiana*[60]*ABHYDROLASE-3*SIGS, HIGS*B. napus*; *A. thaliana*[102,103,104]*SsMNO1*SIGS*B. napus*, *A. thaliana*[112]*SsGAP1*HIGS*N. benthamiana*, *A. thaliana*[100]*SsRAS1/SsRAS2*HIGS*N. benthamiana*, *A. thaliana*[100]*SsOah1*HIGS*A. thaliana*, *Soybean*[97,98,113]*SsPG1*HIGS, SIGS*B. napus*, *N. benthamiana*[113]*SsCBH*HIGS, SIGS*B. napus*, *N. benthamiana*[113]*SsCnd1*HIGS*N. benthamiana*, *A. thaliana,*[99]*SsVPS51/SsDCTN1/SsSAC1*SIGS*Lactuca sativa* var. *ramosa* Hort, *Brassica oleracea* var. *acephala* DC.[20]*SsPac1/SsSmk1*SIGS*B. juncea*, *A. thaliana*[101]*SsAgo2*SIGS*N. benthamiana*[114]


## 5. Challenges of Trans-Kingdom RNAi in the Control of *S. sclerotiorum*

Compared with other plant disease control methods, trans-kingdom RNAi strategies have several potentials in the resistance to *S. sclerotiorum*. Firstly, taking HIGS as an example, it provides an effective crop protection strategy that can replace expensive and environmentally harmful chemical treatments. Secondly, in contrast to traditional resistance (R) genes, HIGS-based resistance tends to be more durable. Traditional R gene-mediated resistance is often quickly overcome due to compensatory mutations in the *Avr* genes of the pathogen, while crops lack inherent resistance genes to *S. sclerotiorum* infection. HIGS offers an alternative control approach. Additionally, trans-kingdom RNAi can be used to control multiple diseases in *B. napus*, for instance, by generating sRNA based on the conserved sequence of *SsGAP1* in *S. sclerotiorum*, which provides resistance not only to *S. sclerotiorum* but also to *B. cinerea* [100]. Furthermore, new trans-kingdom RNAi target genes can be easily designed to combat co-evolving pathogens.

Nonetheless, crops containing RNAi transgenes are currently not permitted to enter the market, likely due to the fact that trans-kingdom RNAi strategies like HIGS requires genetic modification as an auxiliary tool [115]. The strict legislative regulations on genetically modified organisms (GMOs) have delayed the promotion and application of HIGS-based crop improvement strategies. Despite these challenges, in 2017, the U.S. Environmental Protection Agency (EPA) approved a genetically modified corn expressing dsRNA targeting the Western corn rootworm [116]. This transgenic corn, known as SmartStax Pro, is expected to be commercially available in the United States in the coming years. Other species, such as papaya-expressing dsRNA against papaya ringspot virus (PRSV) targeting the coat protein (CP) and helper component proteinase (HC-Pro), as well as plum varieties expressing hairpin RNA molecules targeting the viral CP gene to achieve PPV resistance, have also been approved and cultivated in the United States, Canada, and Brazil [84,117]. With the acceptance of GM technology by the public, it is expected that new opportunities will emerge in the near future. At the same time, the discovery of rootstock–scion communication channels for RNA indicates that grafting is also a promising non-GMO method for dsRNA transfer and expression. By grafting transgenic rootstocks that express dsRNA/hairpin RNA, the target mRNA molecules in the scion can be successfully silenced. This technology has been applied in cherries, and the grafting of RNAi donor plants may serve as a potential tool for bypassing GMO regulations in grafted breeding plants [118,119,120].

Off-target effects represent another challenge for trans-kingdom RNAi technology. Studies have indicated that approximately 50–70% of mRNA transcription in plants may have off-target absorption efficiency [84,85]. This could not only lead to the failure of trans-kingdom RNAi but also affect other aspects of plant growth and development, ultimately impacting crop yield. Therefore, one research has utilized online assessment tools and siRNA scanning tools to calculate and predict off-target effects, helping to select and construct target gene sequences and silencing vectors with low off-target rates [121]. Additionally, studies have shown that exosome-like extracellular vesicles are involved in the trans-kingdom transport of siRNAs [122]. However, current knowledge in this area remains limited, and it is not yet clear whether all plant–microbe interactions rely on the same mechanisms. Further research into the fundamental mechanisms of trans-kingdom RNA silencing is essential to simplify its application in disease control.

Among trans-kingdom RNAi strategies, as a non-transgenic method, SIGS may be more readily accepted by the public compared to HIGS. Moreover, since SIGS targets specific genes, it reduces off-target effects and allows for the customization of sprays targeting specific pests or pathogens to enhance specificity. However, the persistence and the uptake efficiency of exogenously applied dsRNA and siRNA in natural environments is a critical factor affecting the efficacy of trans-kingdom RNA interference [20]. Environmental factors, such as rainfall, ultraviolet radiation, and naturally occurring degradative agents, can inactivate or degrade dsRNA and siRNA exposed to the air. Generally, the effective duration of applied exogenous dsRNA and siRNA under natural conditions ranges from approximately five to seven days. Besides, though studies showed that different lengths of dsRNA and siRNA can be absorbed by pathogens, the shorter dsRNA showed higher efficiency [85,123].

To address this challenge, researchers have developed various strategies to protect naked dsRNA and siRNA. Encapsulating dsRNA in nanoparticles, such as clay, chitosan, liposomes, carbon, gold, and silica [124], or loading RNAi-inducing dsRNA onto layered double hydroxide (LDH) clay nanosheets can effectively extend the functional longevity of exogenous dsRNA and siRNA [125]. Additionally, studies have shown that carbon dot (CDs) and liposome delivery methods can significantly enhance the efficiency of dsRNA and siRNA uptake [84]; however, these methods are costly and involve complex synthesis processes. Regarding these issues, recent studies have discovered the phenomenon of sRNA transmission between fungi and the resulting interspecies silencing among rhizosphere microbes. Based on this, engineered microbes have been developed to induce RNAi in pathogens. For example, engineered *Trichoderma harzianum* produces sRNA to regulate the growth and development of *V. dahliae* and *Fusarium oxysporum* [126]. This RNAi method, on the one hand, takes advantage of the rapid replication characteristic of bacterial microbes to enable large-scale production of dsRNA; on the other hand, due to the specificity of the silencing induced by bacterial microbes [127,128], it offers higher biosafety and controllability compared to VIGS. Additionally, compared to exogenous spraying of naked dsRNA, microbes provide better protection for the produced sRNA, avoiding the potential ecological impacts of nanoparticles [126]. Therefore, this trans-kingdom RNAi pathway, known as MIGS, combines the dual advantages of SIGS and VIGS, making it a future direction to enhance the productivity and efficiency of trans-border RNAi.

Uptake efficiency is another crucial criterion for the efficiency of exogenously applied trans-kingdom RNAi. However, the details of exogenous RNA uptake by pathogens still remain unclear. Currently, there are only a limited number of studies on the uptake mechanisms of fungal dsRNA and the associated genes [129,130,131]. A commonly recognized potential pathway for dsRNA uptake is endocytosis. Study have revealed that inhibiting clathrin-dependent endocytosis significantly reduced dsRNA uptake in *Tribolium castaneum* and weakened RNAi efficiency [132]. Consistent with this finding, silencing clathrin-mediated endocytosis in *S. sclerotiorum* has also been shown to reduce target gene silencing, with six potential regulatory genes—*SsCHC*, *SsAP2*, *SsArf72A*, *SsFCHO1*, *SsAmph*, and *SsVATPase*—being identified in this pathway [130]. Additionally, other studies have indicated that the simultaneous silencing of two putative dsRNA transporter genes, systemic interference defective 1 (*sid-1*) homologs A and C (*SilA* and *SilC*), also reduces target gene silencing [133], suggesting that dsRNA may also be absorbed through a channel-based mechanism.

Since dsRNA uptake and transport are critical steps in the trans-kingdom RNAi process and directly influence its efficiency, yet the specific mechanisms and related genes remain largely unexplored, this area represents a key research direction for overcoming the efficiency bottleneck of trans-kingdom RNAi in the future.

Overall, trans-kingdom RNAi, as a uniquely advantageous method for pest and disease control, has significant potential for improvement and development in terms of yield, efficiency, and transgenic ethics.

## 6. Perspective

So far, chemical methods remain the most direct and effective approach to control SSR in crops. However, growing public concerns over the overuse of chemical agents and the widespread emergence of fungicide-resistant strains have led to an urgent need for alternative methods. Trans-kingdom RNAi technology not only provides a new approach for gene function research but also offers a new environmentally friendly tool for the control of sclerotinia disease. However, it is worth noting that while RNAi technology is highly effective in fungi, there are still risks of exposure to non-target organisms (NTOs) due to factors such as the specificity limitations of target genes, the conservation of the trans-kingdom RNAi mechanism in eukaryotes, and the use of viral vectors. Besides, although there is currently no evidence suggesting that fungi can develop resistance to trans-kingdom RNAi, the possibility of such resistance emerging after large-scale use of RNAi-based biopesticides cannot be ruled out. Therefore, to prevent these issues, we need to systematically conduct environmental risk assessments and resistance risks assessments under field conditions in future research. With the continuous availability of plant and pathogen genome sequences, designing trans-kingdom RNAi targeting specific genes will become easier and more accurate. Thus, using bioinformatics tools to optimize target selection and developing advanced predictive models to assess the impact of RNA on non-target organisms, thereby minimizing off-target effects, will be a key direction in future crop disease control research.

Moreover, other gene-editing tools, such as CRISPR/Cas9, have demonstrated unique advantages in the improvement of plant disease resistance in combination with trans-kingdom RNAi. By targeting and modifying host disease-resistance-related genes (e.g., editing resistance gene promoters or coding regions), CRISPR/Cas9 enables the genetic stabilization of desired traits, making it indispensable in the genetic regulation of crop disease resistance [134,135]. To date, the development of the CRISPR-Cas13 system (e.g., Cas13a/d) has further expanded the functional boundaries of this technology platform. As an RNA-guided RNA nuclease, Cas13 can achieve programmable target recognition by designing specific crRNAs and mediate post-transcriptional gene silencing through the cleavage of target RNA molecules (e.g., plant virus genomes or pest mRNA), exhibiting functional synergy with RNA interference (RNAi) [136].

Thus, due to its multi-target regulatory capability and stability of its delivery system to environmental conditions, trans-kingdom RNAi technology has demonstrated to suppress key pathogenic factors (e.g., toxin synthases or effector protein-encoding genes) of pathogens by directly delivering exogenous dsRNA, making it particularly suitable for emergency control of sudden pathogen outbreaks in agricultural production. From an integrated technology perspective, CRISPR and trans-kingdom RNAi exhibit functional complementarity in plant disease management systems: the former establishes long-term resistance by enhancing the host’s innate immune system through genetic modification, while the latter enables rapid phenotypic control via direct pathogen targeting. Their synergy provides a multi-layered strategy for disease prevention and control, driving the advancement of modern sustainable agriculture.

## Figures and Tables

**Figure 1 pathogens-14-00398-f001:**
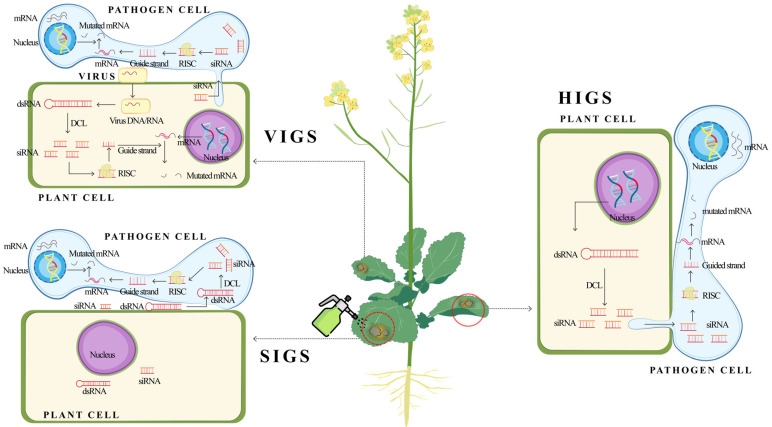
Schematic diagram of three main forms of trans-kingdom RNA interference. HIGS utilizes transgenic technology to express and process dsRNA within the host, delivering dsRNA and siRNA into pathogen cells, where siRNA is unwound and guided to the target gene, inducing gene silencing. SIGS, on the other hand, involves the external application of synthetic dsRNA and siRNA, such as the surface spraying of host plants. Both pathogens and host cells can directly uptake the exogenous dsRNA and siRNA, with pathogens processing dsRNA through their own RNA interference mechanisms. VIGS, however, utilizes the natural defense mechanisms of plants and pathogens against viruses. By delivering siRNA to plant and pathogen cells through virus-mediated gene editing, the endogenous RNAi in both plants and pathogens was induced.

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
