# Peer review of "Trans-Kingdom sRNA Silencing in Sclerotinia sclerotiorum for Crop Fungal Disease Management"

_pathogens, 2025, doi:10.3390/pathogens14040398_

Round 1

Reviewer 1 Report

Comments and Suggestions for Authors

Dear Authors,

                   I  enjoyed reading your review article titled “Trans-kingdom sRNA Silencing in Sclerotinia sclerotiorum for Crop Fungal Disease Management”. The current problems with S. sclerotium disease control due to the lack of resistant germplasm and excessive dependency on chemical control which are detrimental to the environment are alarming. The review focuses on relatively safer and eco-friendly methods of control by the use of Trans-kingdom sRNA silencing-based technologies such as SIGS, HIGS, and VIGS as tools for controlling Sclerotinia sclerotiorum. Your review is comprehensive, well-informed, and highly relevant to researchers focused on S. sclerotiorum research and management. Each section is well-structured, with up-to-date references that strengthen the scientific foundation of your insights. Your writing is clear and engaging, making complex concepts easily understandable to the audience while maintaining scientific rigor. I particularly appreciate how you have synthesized recent advancements in Rnai-based control of diseases in plants and the bottlenecks in the implementation of the technology. I was impressed to see the mention of RNA uptake mechanism present in the host. However, I would love to see the references in the study demonstrating the genes important for RNA uptake and transport. You have mentioned that with the advancement of CRISPR tools the trans-kingdom RNAi is expected to become a crucial tool in controlling SSR can you elaborate on this? There is a mention of CRISPR in the abstract and at the end nothing is mentioned- elsewhere. I have only a few minor suggestions for refinement, which are in the comments in the PDF file of the review article attached. Overall, in my opinion, this is an excellent review and a valuable contribution to the field. Thank you for your effort in putting together an insightful review.

Thank you

Author Response

We appreciate the reviewers for their valuable comments and suggestions, which certainly help us to improve the quality of our manuscript. All comments are addressed on a point-by-point basis below, where letters C & R denote Comment and Response, respectively. The modifications of the text, figures and tables have been highlighted.

C1:Does author mean all types of viruses? DNA/RNA (L52)
R: According to literatures and research, both DNA virus and RNA virus can be used as vector in
VIGS, for instance bean pod mottle virus and Tobacco rattle virus is the widely used RNA virus,
and tomato golden mosaic virus (TGMV), cabbage leaf curl virus (CbLCV) and african cassava
mosaic virus (ACMV) are the commonly used DNA virus for VIGS. However, The VIGS system
based on tobacco rattle virus (TRV) is currently the most widely used system, probably because of
its wide range of hosts.
[Reference]
1. Liu Y, Lyu R, Singleton JJ, Patra B, Pattanaik S, Yuan L. A Cotyledon-based Virus-Induced
Gene Silencing (Cotyledon-VIGS) approach to study specialized metabolism in medicinal
plants. Plant Methods. 2024 Feb 12;20(1):26. doi: 10.1186/s13007-024-01154-x.
2. Jagram N, Dasgupta I. Principles and practice of virus induced gene silencing for functional
genomics in plants. Virus Genes. 2023 Apr;59(2):173-187. doi: 10.1007/s11262-022-01941-5.
C2: I would suggest better resolution figure. Some of the characters in the figure are cluttered and
cannot be read without 150% magnification (L201)
R: Thanks for your suggestion! The figure is adjusted and replaced.
C3: Expressing what? Do you mean plant expression vector expressing dsRNA? Maybe
restructure the sentence so that it becomes clearer (L215)
R: Thanks for your comments! The word “expressing” was misused, and is deleted.
C4: I would suggest author to start this paragraph with the challenges e.g. regulatory issues that
must be overcome before RNAi-mediated transgenic plants can be trailed in the field (L226)
R: Relevant content is added (L226-228).
Q5:Can author add a few sentences on VIGS. e.g. Mechanistically VIGS is the use of engineered
plant viruses to directly inoculate plants or use vector to transform plant cells
with viral genome etc.(L241)
R: Relevant content is added (L241-243).
Q6: This paper talks about Botrytis and Verticillium and does not mention about S. sclerotium
effectively taking up environmental dsRNA. (L284)
R: Thanks for your comments! The reference is replaced.
C7:Very well written. I would suggest author to write a few sentences around the applicability of
HIGS in S. sclerotium since the pathogen spend significant time as necrotrophs. would the HIGS
work? (L330)
R: As mentioned in the text, Sclerotinia sclerotiorum is not a purely necrotrophic pathogen. It has
a transient biotrophic phase, during which its growth and development, as well as its preparation
for host infection, are important factors contributing to its virulence. Currently, trans-kingdom
RNAi has been used to regulate genes associated with this stage in S. sclerotiorum, such as SsSnf5,
SsSte50, SsCnd1, and SsCak1, which are involved in the regulation of the infection cushion,
appressorium formation, and maturation during the early infection stages. In addition,
trans-kingdom RNAi also affects the virulence of S. sclerotiorum during its necrotrophic stage by
regulating key virulence factors responsible for killing host plant cells, such as OA (oxalic acid)
and ROS (reactive oxygen species), through genes like SsOah1, SsTrx1, SsPG1, and SsCBH.
The relevant content has been added to the text (L258-262).
C8:Do we call it a green biological control? It‘s categorized as a transgenic and are not accepted
by majority of the countries. (L338)
R: Indeed, while HIGS as a transgenic technology avoids environmental pollution caused by
chemical agents, it does have certain ethical controversies and biosafety concerns. Therefore,
describing it as "safe" and "green" is not entirely accurate, so the expression is adjusted (L338).
C9: Suggest references in the study demonstrating the genes important for RNA uptake and
transport.
R: Thanks for your comments! Relevant content and references are added (L408-425).
C10: More detail about the advancement of CRISPR tools the trans-kingdom RNAi.
R: Relevant discussion is added (L437-460).

Reviewer 2 Report

Comments and Suggestions for Authors

Overall Summary: Ouyang et al. present a review article that focuses on the evaluation of trans-kingdom RNA silencing and its use in S. sclerotiorum disease management. The review gives an overview of genes involved in Sclerotinia virulence, followed by an overview of RNAi processes and the applications in the S. sclerotiorum pathosystem. The review does a good job of summarizing relevant information in these subtopics. Suggestions on improving the review through some modification of data visualization and added information about virulence genes and CRISPRs is mentioned below.

Specific comments:

  • Section 2: Since this section is focused on virulence factors that are recently discovered in the past 5 years or so, it would be good to add a short paragraph early in this section to highlight the main findings before 2020 or so that were the primary known virulence genes in Sclerotinia. The subsequent sections can then also cover how the gaps in these previous findings were filled by the newer data.
  • Table 1 could be better represented as the main table plus an added figure where the virulence factors are depicted in terms of their dependence on other genes. For instance, StuA and Smk3; Os4 and glpA. This would make it easier to understand the relationship between these genes as the list presented is quite long.
  • To inform future efforts on gene homology mining, it would also be useful to mention the whole genome sequencing efforts for S. sclerotiorum and the assembled genomes/accessions available for bioinformatics.
  • While the review does mention in the introduction about CRISPR/Cas gene editing to deploy RNAi, there isn’t much of a discussion about this. This can be included as a section in the manuscript where there is discussion about the editing platforms and their use in RNAi in field crops overall. This could also include the variety of new Cas editing systems that are being used currently.

Author Response

We appreciate the reviewers for their valuable comments and suggestions, which certainly help us to improve the quality of our manuscript. All comments are addressed on a point-by-point basis below, where letters C & R denote Comment and Response, respectively. The modifications of the text, figures and tables have been highlighted.

C1: Section 2: Since this section is focused on virulence factors that are recently discovered in the 
past 5 years or so, it would be good to add a short paragraph early in this section to highlight the 
main findings before 2020 or so that were the primary known virulence genes in Sclerotinia. The 
subsequent sections can then also cover how the gaps in these previous findings were filled by the 
newer data.
R: Thanks for your comments! Cases and how newest data fill the gaps based on findings before 
2020 are added (L68-77).
C2: Table 1 could be better represented as the main table plus an added figure where the virulence 
factors are depicted in terms of their dependence on other genes. For instance, StuA and Smk3; 
Os4 and glpA. This would make it easier to understand the relationship between these genes as the 
list presented is quite long.
A: Thank you for your comments! We appreciate your input and totally agree with you. However, 
most of the genes listed in the table function independently and might not have direct connections 
that can be effectively represented in a single figure. The "Interacting Genes/Proteins" column is 
primarily intended to provide a quick overview of how these virulence genes exert their functions, 
their roles in the growth and pathogenic mechanisms of Sclerotinia sclerotiorum, and their 
interactions with other genes, offering context for their significance in the pathogen’s physiology 
and virulence.
While the information of interactions may be useful, the "Interacting Genes/Proteins" column
make the list longer. However, deleting this column to short the list, and depicting this information 
in a figure could introduce unnecessary complexity and potentially obscure key insights.
Consequently, we have not yet found a better way to present these independent and fragmented 
pieces of information regarding gene interactions. Therefore, we have temporarily retained the 
table format.
C3: To inform future efforts on gene homology mining, it would also be useful to mention the 
whole genome sequencing efforts for S. sclerotiorum and the assembled genomes/accessions 
available for bioinformatics.
R: Thanks for your comments! Relevant content is added (L183-194).
C4: While the review does mention in the introduction about CRISPR/Cas gene editing to deploy 
RNAi, there isn’t much of a discussion about this. This can be included as a section in the 
manuscript where there is discussion about the editing platforms and their use in RNAi in field 
crops overall. This could also include the variety of new Cas editing systems that are being used 
currently.
R: Thanks for your comments! Relevant discussion is added (L437-460)

Reviewer 3 Report

Comments and Suggestions for Authors

The review article summarizes potential and already demonstrated cross-kingdom RNAi targets from characterized virulence-related genes of S. sclerotiorum. Overall, it looks pretty comprehensive and I just have a minor suggestion that the authors use either cross- or trans- but don't use them both interchangeably. 

Another minor suggestion is that the authors should discuss whether these cross-kingdom RNAi strategies may or may not induce fungi to develop fungicide resistance to these RNAi fungicides. Insects have found to develop resistance.

Also, the authors should discuss ways to track these RNAi pesticides to ensure that they don't target other harmless organisms in nature once deployed in the field.

Author Response

We appreciate the reviewers for their valuable comments, which certainly help us to improve the quality of our manuscript. All comments are addressed on a point-by-point basis below, where letters C & R denote Comment and Response, respectively. The modifications of the text, figures and tables have been highlighted.

Reviewer 4 Report

Comments and Suggestions for Authors

This review summarizes recent advances in the SCLEROTINIA SCLEROTIORUM-HOST PLANT pathosystem. The authors review recent data on the fungal pathogenic genes, the effectors and their targets. They summarize knowledge on the phenomenon of cross kingdom RNAi inhibition in natural and genetic manipulated SYSTEMS  and the applications of this technique through host induced gene silencing (HIGS) or spray induced gene silencing (SIGS). Since natural resistance to Sclerotinia is polygenic and not complete for most of the crop plants, development of alternative methods through biotechnology is very important to combat this broad host range, world-wide present pathogen.   

Below follow some comments on the manuscript. The lines are indicated.  

----------------------
line 172
with plant defense mechanisms via (micRNA)  miRNA-mediated gene regulatory networks.

miRNA instead of micRNA ?

----------------------------------------------------------------------------------------------
lines 231-232
formulations targeting key pathogen genes (e.g., DCL1/DCL2) can effectively suppress the development...

dcl1 and dcl2 are always key pathogen genes? add information of how many plant (pathogenic or not) fungal genomes 
contain components of the rnai machinery. are other effector proteins not more suitable to be targeted?

----------------------------------------------------------------------
lines 240-241
uses engineered viruses or pant cells containing meditated viral genome to evoke RNAi in host 

instead of meditated , manipulated or genetically engineered (?)

----------------------------------------------------------------------------
line 243

.. such as RNA transcript rub inoculation, the virus carrying the constructed target gene

In general, rna transcript rub inoculation of viral genomes is in most cases an ineffective or even not succesful  way of infection (with the exception of viroid inocculation). Effective infection is achieved either with fresh sap from infected plants containing intact (engineered or wild type) virus particles (only for mechanically transmitted plant viruses), or by agroinfection of the  viral genome (a technique which  however, should not be used in non contolled environment). To my opinion, vigs is useful only in laboratory settings.

----------------------------------------------------------------------
line 299
To date, there have been no report on cross-kingdom RNA interference directly in...

There have been no reports on cross kingdom ....

----------------------------------------------------------------------

Line 447-448
Thus, due to its multi-target regulatory capability, and adaptability of its delivery system to environmental conditions, cross-kingdom RNA interference (RNAi) technology... 

..adaptability of its delivery system to environmental conditions...
Explain why cross kingdom rnai is adaptable to environmental conditions (?)
-----------------------------------------

Author Response

(The authors gave the same response as above.)
